# High-Resolution Photoemission Study of Neutron-Induced Defects in Amorphous Hydrogenated Silicon Devices

**DOI:** 10.3390/nano12193466

**Published:** 2022-10-04

**Authors:** Francesca Peverini, Marco Bizzarri, Maurizio Boscardin, Lucio Calcagnile, Mirco Caprai, Anna Paola Caricato, Giuseppe Antonio Pablo Cirrone, Michele Crivellari, Giacomo Cuttone, Sylvain Dunand, Livio Fanò, Benedetta Gianfelici, Omar Hammad, Maria Ionica, Keida Kanxheri, Matthew Large, Giuseppe Maruccio, Mauro Menichelli, Anna Grazia Monteduro, Francesco Moscatelli, Arianna Morozzi, Stefania Pallotta, Andrea Papi, Daniele Passeri, Marco Petasecca, Giada Petringa, Igor Pis, Gianluca Quarta, Silvia Rizzato, Alessandro Rossi, Giulia Rossi, Andrea Scorzoni, Cristian Soncini, Leonello Servoli, Silvia Tacchi, Cinzia Talamonti, Giovanni Verzellesi, Nicolas Wyrsch, Nicola Zema, Maddalena Pedio

**Affiliations:** 1INFN (Istituto Nazionale di Fisica Nucleare), Sez. di Perugia, Via Pascoli s.n.c., 06123 Perugia, Italy; 2Dipartimento di Fisica e Geologia, Università degli Studi di Perugia, Via Pascoli s.n.c., 06123 Perugia, Italy; 3INFN (Istituto Nazionale di Fisica Nucleare), TIPFA (Trento Institute for Fundamental Physics and Applications), Via Sommarive 14, 38123 Trento, Italy; 4Fondazione Bruno Kessler, Via Sommarive 18, 38123 Trento, Italy; 5CEDAD-Centro di Fisica Applicata, Datazione e Diagnostica, Dipartimento di Matematica e Fisica “Ennio de Giorgi”, Università del Salento e INFN-Sezione di Lecce, 73100 Lecce, Italy; 6INFN (Istituto Nazionale di Fisica Nucleare) and Dipartimento di Fisica e Matematica dell’Università del Salento, Via per Arnesano, 73100 Lecce, Italy; 7INFN (Istituto Nazionale di Fisica Nucleare) Laboratori Nazionali del Sud, Via S. Sofia 62, 95123 Catania, Italy; 8Ecole Polytechnique Fédérale de Lausanne (EPFL), Institute of Electrical and Microengineering (IME), Rue de la Maladière 71b, 2000 Neuchâtel, Switzerland; 9Centre for Medical Radiation Physics, University of Wollongong, Northfields Ave, Wollongong, NSW 2522, Australia; 10Istituto Officina dei Materiali-CNR, Basovizza SS-14, km 163.5, 34012 Trieste, Italy; 11INFN (Istituto Nazionale di Fisica Nucleare) and Dipartimento di Fisica Scienze Biomediche Sperimentali e Cliniche “Mario Serio”, Viale Morgagni 50, 50135 Firenze, Italy; 12Dipartimento di Ingegneria, Università Degli Studi di Perugia, via G.Duranti, 06125 Perugia, Italy; 13IOM-CNR, Istituto Officina dei Materiali, AREA Science Park Basovizza, 34149 Trieste, Italy; 14Dipartimento di Scienze e Metodi dell’Ingegneria, Università di Modena e Reggio Emilia, Via Amendola 2, 42122 Reggio Emilia, Italy; 15Istituto di Struttura della Materia-CNR, Via Fosso del Cavaliere 100, 00133 Roma, Italy; 16Istituto Officina dei Materiali-CNR, Via Pascoli s.n.c., 06123 Perugia, Italy

**Keywords:** amorphous hydrogenated silicon, photoemission, X-ray absorption

## Abstract

In this paper, by means of high-resolution photoemission, soft X-ray absorption and atomic force microscopy, we investigate, for the first time, the mechanisms of damaging, induced by neutron source, and recovering (after annealing) of p-i-n detector devices based on hydrogenated amorphous silicon (a-Si:H). This investigation will be performed by mean of high-resolution photoemission, soft X-Ray absorption and atomic force microscopy. Due to dangling bonds, the amorphous silicon is a highly defective material. However, by hydrogenation it is possible to reduce the density of the defect by several orders of magnitude, using hydrogenation and this will allow its usage in radiation detector devices. The investigation of the damage induced by exposure to high energy irradiation and its microscopic origin is fundamental since the amount of defects determine the electronic properties of the a-Si:H. The comparison of the spectroscopic results on bare and irradiated samples shows an increased degree of disorder and a strong reduction of the Si-H bonds after irradiation. After annealing we observe a partial recovering of the Si-H bonds, reducing the disorder in the Si (possibly due to the lowering of the radiation-induced dangling bonds). Moreover, effects in the uppermost coating are also observed by spectroscopies.

## 1. Introduction

There is a growing need for radiation-resistant detectors capable of high dynamic range and precise flux measurements for a wide variety of particles and ionizing radiation. The development of new accelerator techniques requires the development of hard beam monitoring detectors for dosimetry and new detector able to make flux measurement.

The main challenge is to find sensitive materials that simultaneously possess a large dynamic range, a thin substrate, radiation-resistance, and the ability to function both in air and vacuum. Hydrogenated amorphous silicon (a-Si:H) is an optimal candidate due to its bandgap (wider than crystalline Silicon (c-Si)) and its intrinsic non-crystalline nature that offers a natural resistance to radiation-induced displacement damages. Moreover, the possibility to grow it as very thin layer and its mature fabrication technology do to its use for solar cells makes it advantageous in terms of production costs.

Hydrogenated amorphous silicon (a-Si:H) was first synthesized by radiofrequency Plasma Enhanced Chemical Vapor Deposition (PECVD) from a mixture of silane gas (SiH4) and hydrogen [1] in 1969. This material was doped by insertion of Diborane (p-type) and Phosphine (n-type) in the gas mixture [2], and consequently used for the construction of electronics, solar cells and radiation detectors. First studies in the framework of solar cell production for space applications highlighted its optimal radiation hardness of this material [3].

However, the irregular lattice structure of the amorphous silicon induces a high concentration of defects (that can reach densities of 1019 cm^−3^), strongly limiting the use of such materials in electronic devices. To overcome such limitation, hydrogen was added during the deposition process (PECVD from a mixture of silane gas and hydrogen gas). The hydrogenation saturates most of the dangling bonds, lowering the density of the defects to 10^15^ cm^−3^. The typical amount of hydrogen required to obtain a-Si:H quality for detector applications is in the order of 10% atomic hydrogen [4] The inclusion of hydrogen has the additional effect of increasing the band gap to 1.7–1.9 eV [5].

The high radiation hardness of a-Si:H was recognized very early, and this property was one of the driving forces behind the use of this material as a radiation detector. Several irradiation tests were performed using different radiation source such as proton [3,6], gamma [7], neutron [8], or heavy ions irradiation [9] on thin or thick p-i-n devices.

In all studies the effect of irradiation damage on the behavior of the sensors has been investigated, and a comparison of electric characteristics measured prior and after the irradiation and the same measurements performed after annealing highlight a decrease in performance, due to irradiation, and recovery after annealing. However, a fundamental characterization of the microscopic properties associated with the variation of a-Si:H device responses following irradiations is missing in the literature. Clarify such effects and get a deeper understanding of the mechanisms underpinning the macroscopic variations observed are critical issues to address in order to optimize the a-Si:H applications.

In this paper, we use High-Resolution X-ray photoemission (XPS) and soft X-ray absorption (XAS) [10] to confirm the results and study at a more fundamental level the impact of neutron irradiation and subsequent annealing on a-Si:H p-i-n detectors previously characterized by electric measurements [11]. All measurements were performed on a-Si:H device prototypes in order to characterize the behavior in real systems. We obtained similar results to ref. [6] in which the displacement damage radiation was studied by using protons. The total dose obtained with neutrons of this study is comparable, in the NIEL approximation, to the displacement dose of ref. [6]. The present level of neutron irradiation is related to LHC experiment, with refer to ATLAS and CMS experiments central detectors in several years of operation with present LHC fluxes.

Previous characterization of a-Si:H films by photoemission and XAS are reported in the literature [12,13,14]. However, due to the low instrumental resolution it was not possible disentangle the different contributions of inequivalent Si atoms in the spectra (Si-H and amorphous Si-Si bonds).

Nowadays, due to instrumental improvements, the spectroscopic fingerprint of the hydrogenation and the contribution of the different inequivalent Si atoms are easily distinguishable. Nevertheless, to the best of our knowledge, the only study in the literature which report a spectroscopic characterization of a-Si:H systems uses high resolution hard X-ray XPS in the 3, 5, 8 KeV ranges [15]. Here, we successfully characterize a-Si:H device prototypes by using high resolution XPS in the soft X photon energy with a twofold aim: define a protocol for XPS and XAS measurements in characterizing future real devices and get a deeper understanding of the damaging mechanism induced by irradiation. Here, we first report of neutron source irradiation, but studies of gamma ray and protons induced damaging are planned in the next future.

Core level photoemission [16] is a surface-sensitive technique allowing to single out the different inequivalent Si atoms in the sample. This technique was chosen because is sensitive to the surrounding distribution of inequivalent Si atoms: the chemical shift of the core levels reflects the different bonding of the studied element. By accurate fit of the Si2p core level into its components, it is possible to quantitatively estimate the number of Si-H bonds. Moreover, the energy shift of the Si2p photoemission features allowing to distinguish between Si atoms in the oxide coating from that one in the underneath film.

To get information on the effect of the irradiation in the bulk XAS was used. During XAS measurements core-holes are generated, which decay through competitive processes as Auger electron emission and fluorescence radiation, proportionally to the absorption cross-section. At L-edge photon energies, in light elements (such as Si, P, S), Auger and Total Electron yield (obtained for example by measuring the drain current of the sample) exceeds the fluorescence yield (FY) by several orders of magnitude [17]. However, the sampling depths considerably differs in the two cases, few tenths of nm and hundreds of nm, respectively. All Si L_2,3_ XAS measurements here reported were performed FY mode (sampling depth of about 60–80 nm), overcoming the limit of the surface-sensitivity of XPS [18].

The comparison of the spectroscopic results on the different a-Si:H samples shows that irradiation induces an increased degree of disorder and a strong reduction of the amount of the Si-H bonds (Section 3.1.1 and Section 3.1.2). The subsequent annealing produces a reduction in the amount of the disordered Si with a possible reduction of the number of dangling bonds, this can be correlated with the capability of the annealing process to partially compensate for the damage due to neutrons as shown in ref. [11]. Moreover, effects on the uppermost coating are also observed and described in Section 3.2.

## 2. Materials and Methods

The a-Si:H films were deposited on support wafers (typical a Cz low resistivity p-type silicon wafer with a resistivity below 10 Ωcm) via Plasma Enhanced Chemical Vapor Deposition (PECVD) with a very high frequency (VHF) excited plasma at the frequency of 70 MHz. a-Si:H films were deposited at IME laboratory EPFL (Ecole Polytechnique Federale De Lausanne), Neuchâtel (Switzerland) and the p-i-n devices (Figure 1) were fabricated at FBK (Fondazione Bruno Kesseler) Trento.

The irradiation of the detector, p-i-n strip device (10 µm thick 0.2 × 5 mm^2^ surface) has been done with neutron at the JSI (Jozef Stefan Institute) reactor in Ljubljana up to the total fluence of 1016 neq/cm^2^, to investigate the damage generated by the radiation. As in any nuclear reactor there is always gamma radiation background, which can usually represent about one-third of the total dose received by the sample, in our case for an estimation given by the facility staff the gamma rays dose rate expected is about 50 Gy/s that for a total irradiation time of 2625 s is about 130 kGy [19]. The device response to X-Ray irradiation versus dose rate was taken before and after neutron irradiation and after 12 h of annealing at 100 °C, to determine the dosimetric sensitivity of 30 kV X-rays, which was calculated as the slope of the current versus dose rate straight line.

The linearity was very good under all conditions and the sensitivity changed from 2.39 nC/cGy for the non-irrdiated component to 1.13 nC/cGy after irradiation; after the 12 h of annealing the sensitivity increased up to 3.0 nC/cGy.

Additionally, the trend of leakage current suggests the same kind of behavior, indeed compared to the pre-irradiation values an increment of leakage current can be observed after irradiation. After 12 h of annealing, a new measurement was performed, and leakage current returned to pre-irradiation values [11].

Similar effects are found in reference [6] where a 32.6 µm diode was irradiated at CERN IRRAD1 facility with 24 GeV protons at the rate of 3 × 1013 p/(h × cm^2^) up to the total fluence of 1016 p/cm^2^.

For the spectroscopic characterization the a-Si:H devices (not irradiated a-SiH NonIrr, irradiated a-SiH Irr, and annealed a-SiH Ann) have been exfoliated in air employing adhesive tape to remove the top metallic layer, and the exfoliated samples were characterized by in air Atomic Force Microscopy (AFM) and photoemission measurements in Ultra High Vacuum (UHV). The reference sample (a-SiH Ref) is an a-Si:H film with 10 nm SiO_2_ coating.

The SiO_2_ coating of the a-SiH Ref was partially removed by Ar+ sputtering (1 KeV Ar+ beam, current on sample 1.8 μA) in the UHV chamber, leaving a layer of about 2 + 1 nm, to avoid defects created by the sputtering process in the a-Si:H film.

The not-hydrogenated amorphous silicon sample (a Si) shown in Figure 2 was deposited by resistive heating of an undoped Si wafer onto an Indium Tin Oxide (ITO) substrate.

Photoemission measurements were performed at the Circular Polarization (CiPo) and BACH beamlines (ELETTRA, Trieste). All the photoemission binding energies are referenced to the Fermi level and Au 4f core levels measured on a polycrystalline Au in electric contact with the samples.

At the CiPo beamline [20] photoemission data were taken using a grazing incidence monochromator (SGM) and the photoelectron spectra were acquired by using an OMICRON 125 hemispherical electron energy analyzer collecting electrons at normal emission. The overall energy resolution at 650 eV photon energy was about 1.2 eV.

BACH beamline [21] is equipped with a hemispherical electron energy analyzer (Scienta R3000) at an angle of 60° with respect to the X-ray incidence direction. Core level spectra were acquired with 700 eV photon energy and 220 meV energy resolution and at 1400 eV photon energy and 700 meV energy resolution. All spectra were recorded at normal emission.

The probing depths associated with the spectroscopies is a crucial information, this quantity in photoemission is obtained through the IMFP, related to the distance that electrons travel before suffering an inelastic collision and exciting plasmons or vibrations. The IMFP of Si2p electrons by Photoemission is about 1.5 nm (namely 1,9 nm for SiO_2_, 1.4 nm for Si, respectively), and about 3 nm (3.6 nm for SiO_2_, 2.9 for Si), when using, respectively 700 and 1400 eV photon energy, as calculated from [22] (see Appendix A). It is worth noting that the estimation of the detected probe depth for photoemission equals approximately three times the Inelastic Mean Free Path. In this paper, we present the data obtained from the measurement taken at BACH beamline that allowed us to reach higher photon energies, i.e., deeper escape depth from the sample.

Si L_2,3_ (that measure the transition from the Si2p to the empty states) and O K edge (transitions from O1s) X-ray absorption (XAS) spectra were measured in fluorescence yield (FY) using a multi-channel plate (MCP) detector (F4655 Hamamatsu) at BACH beamline. The sampling depth is about 60–80 nm.

Contact mode AFM measurements were carried out in air using a Solver Pro scanning probe microscope (NT-MDT, Moscow, Russia), using rectangular silicon cantilevers, 350 μm long and 35 μm wide.

## 3. Results

### 3.1. Effects on a-Si:H

The a-SiH Ref and device samples were characterized by High-Resolution Si2p Core level photoemission, XAS Si L_2,3_.

#### 3.1.1. Si L_2,3_ XAS

XAS of Si L_2,3_ edges taken in fluorescence mode (FY) are used to enlighten the difference in the Si surrounding at a sampling depth of about 70 nm.

Figure 2 shows the spectra of the *a-SiH Ref* samples taken in this work, compared with Crystalline Si (c-Si) and not-hydrogenated amorphous Si (a-Si) spectra measured in the same experimental condition and the same measure obtained for the other three device samples: not irradiated (*a-SiH NonIrr*), Irradiated (*a-SiH Irr*) and the annealed one (*a-SiH Ann*).

The energy separation between the elementary silicon part (photons energies below 104 eV) and silicon oxides part (photons energies higher than 104 eV) (see Appendix A), accompanied by a well-defined fine structure, allows to single out the changes in local atomic surrounding over the analyzed samples.

The Si L_2,3_ XAS spectra of the c-Si, not-hydrogenated a-Si and hydrogenated amorphous (a-Si:H) lineshape are different reflecting the sensitivity of the technique to the local environment of the silicon atoms. The c-Si shows a sharp edge and a complex fine structure of features, while a-Si presents a tail and two features above the edge [23] at about 100 and 100.5 eV. The a-SiH Ref spectrum is characterized by broader peaks as compared with the c-Si with a smeared group of features at about 100–101 eV photon energy, in agreement with the literature [12]. It is worth noting that the XAS is sensitive to the local surrounding of the excited atom (typically in a range of 1–5 nm, according to the element and the edge) and the spectrum of a non-homogeneous sample includes the contributions of the different surroundings.

Even though the detailed discussion of the features in Si L_2,3_ XAS is beyond the scope of this work, Figure 2 indicates that after irradiation, the lineshape changes, becoming more similar to the a-Si standard with the two features A and B at 100 and 100.5 eV, respectively. The subsequent annealing (*a-SiH Ann* spectrum) recovers the features of the a-*SiH NonIrr* spectrum.

#### 3.1.2. Si2p Core Level Photoemission

High-resolution photoemission is applied to model the non-equivalent surroundings in the a-SiH Ref and device samples. We remind that the Si2p photoemission provides information on the initial core level states while the Si L_2,3_ XAS gives information on the joint density of the transitions between the Si2p initial states and the empty states of the Silicon. See for example [10,22,24].

The a-SiH Ref sample Si2p XPS spectrum in Figure 3 shows a broad emission at binding energy (BE) of about 103–104 eV and a more structured peak at lower BE. In agreement with the literature [12,14,24] the first feature is assigned to the oxide coating topmost oxide layer (SiO_2_ with Si^+4^ oxidation), while the other group of features is assigned to the a-Si:H film. The SiO_2_ overlayer is about 2.5 nm thick, as derived from the relative intensities of Si-O and the substrate peaks in the Si2p spectrum [24]. Our interpretation is confirmed by the observation that the high BE contribution gradually disappears as the sample is continuously sputtered (Appendix A). It is worth noting that the clear energy separation of the two peak groups is crucial to get separate information about the coating and the underneath films, even at the photon energy (1400 eV) used in this work when compared with High photon energy photoemission (HAXPES) measurements performed at 3, 5, 8 KeV [15]. To single out the a-Si:H film components, we performed a fit of the spectra by the KolXPD code [25]. The baseline of the Si2p is fitted with a set of Voigt doublets with Si2p_3/2_-Si2p_1/2_ spin-orbit splitting of 0.6 eV and the gaussian contribution that includes the energy resolution and the disorder. The Si2p_3/2_ components at 99.1 eV, 99.4 eV and 99.7 eV, in agreement with ref. [15], are assigned to the component labeled Dis Si related to the amorphous silicon (including a statistical distribution of bond lengths and dangling bonds), the crystalline c Si-Si and hydrogenated Si-H silicon, respectively. The deconvolution fit parameters are listed in the Appendix A. Because of the presence of a weak C1s peak, found at about 283.4 eV, in the spectra of all four samples, the carbon contamination has been also taken into account adding a new component for Si-C bonding at 100.1 eV.

It is worth noting that the presence of a very weak component (c Si-Si) of crystalline silicon can be related to a weak polymorphism present in a-Si:H films obtained with the plasma growth procedure [26] resulting in a small number of crystalline grains.

The second step was to measure and compare the Si2p spectra in the Si2p devices’ spectra before and after irradiation and annealing. The remaining oxide coating of the three exfoliated samples resulted in about 1–2 nm according to the Si2p depth profiling calibration obtained on the a-SiH Ref as a function of the sputtering time (Appendix A). The Si2p XPS spectra of the non-irradiated sample (a-SiH NonIrr), irradiated (a-SiH Irr) and annealed (a-SiH Ann) are reported in Figure 4. The three samples had no sputtering treatment.

At variance with the a-SiH Ref (Figure 3) the presence of the c Si-Si component increases in the exfoliated samples. Figure 4 compares the Si2p core levels measured on a-SiH NonIr, a-SiH Irr and a-SiH Ann devices taken at 1400 eV photon energy and the a-SiH NonIr at 700 eV. The spectrum at hv = 700 eV photon energy (i.e., better resolution but lower detecting depth) shows this component even better.

We ascribe this to the influence at the topmost part of the a-Si:H film of the removed Al-Cr layer. The process of metal deposition aims to provide electrical contact to the semiconductor device, the thin Cr layer is used to avoid the diffusion of Al into the n-type a-Si:H layer changing the doping level of the layer itself, Cr can also prevent crystallization induced by Al [27,28]. Nevertheless, the crystalline silicon (c Si-Si) component at BE of 99.4 eV results increased at lower detecting depth suggesting that the amorphous Silicon crystallizes below the metal contact during the metallization process.

Table 1 reports the relative intensities of the Si2p components, excluding the oxide layer contributions (fit parameters used for the global fit are reported in Appendix A). By comparing the intensities of the peaks to the total area subtended by the peak group of the Si2p film, i.e., excluding the contribution of the coating, we notice that, after irradiation, dangling bonds contribution (Dis Si) in a-Si:H film increases from 46% to 67% while the relative peak of Si-H bonds decreases with a reduction from 11% to 3%. This trend is in qualitative agreement with Si L_2,3_ XAS measurements. After annealing a recovery of the Si-H concentration takes place up to 12%, with a decrease of the Dis Si component practically to the value of the non-irradiated sample.

We stress that the fractions provided in Table 1 can be taken only as indicative of a trend and not considered absolute. It is worth noting that the XAS Si L_2,3_ sampling depth presented in the previous section is about one order of magnitude higher than these photoemission results. We used different component models to decompose the three spectra of the peak group of the Si2p film, excluding the contribution of the coating, including or not the Carbon contamination of Silicon and additional components of the coating. It is worth noting that the high broadening present in the spectra implies that the “best fit” obtained by minimizing the residual and χ^2^ has a non-unique fit solution. In any case, the trend of the Si-H and a-Si fractions is common to all the different fit models. Errors in the concentration deduced from fit reported in the literature [29] are about 2%; in the present case we evaluate an error of about 4% extimated from the spread of the values obtained by the different deconvolution models (see Table 1).

In summary Si2p core level measurements discriminate the Si configuration within the device before and after neutron irradiation and subsequent annealing. The irradiation induces an increase of the Si dangling bonds within the a-Si:H layer and a decrease in the Si-H bonds. The annealing restores the Si-H component.

### 3.2. Effect of Irradiation on the SiO_2_ Coating

The effect of irradiation and annealing on the coating of the device samples have been characterized by AFM and O1s photoemission, XAS O K edge.

Figure 5 shows the AFM topography and the corresponding height profiles of the non-irradiated and irradiated samples. It is worth noting that the three samples had no sputtering treatment.

As can be seen in Figure 5a before irradiation the sample surface exhibits an RMS roughness of about 3 nm and it is characterized by the presence of rare aggregates, whose height ranges between 1 and 10 nm (Figure 5b). After irradiation, the RMS roughness increases to about 6 nm and the formation of closer and larger aggregates is observed, reaching a height up to about 50 nm (Figure 5c,d). Finally, after the annealing procedure, we observe the formation of larger aggregates, having a height up to about 80 nm, as can be seen in Figure 5e,f.

The in-plane area of defects has been estimated using the software for image analysis *Gwydion* [30] in the three cases. The in-plane defects of the *a-SiH NoIr* have an area density of about 2%, after irradiation, the area of defects results increased to about 12%, while after annealing it is reduced to 6%.

Photoemission and XAS are applied to get complementary information on the effect of irradiation and annealing treatment on the coating. Figure 4 shows that the Si2p core level in the three device samples presents a larger feature of the coating component, with an additional broad peak at the BE of about 102 eV. In particular, the Si2p spectrum of the annealed a-SiH Ann sample shows a shift of the oxidized component at lower BE. This is the indication of the presence of SiOx non-stoichiometric oxides (with Si oxidation state × intermediate between 4 and 0) after device treatments [24]. The O 1s photoemission at hv = 1400 eV and the O K edge XAS measurements are shown in Figure 6.

In the non-irradiated sample case, a single O 1s component is located at 533.4 eV, while in the other two cases the O1s spectra result broader and exhibit two contributions. The deconvolution of the O1s is performed by Voigt functions. The additional peak in the irradiated device, appearing at lower BE, is assigned in the literature to oxygen vacancies in the case of X-ray irradiation of SiO_2_ [31].

In the case of the annealed sample, the maximum of the O1s spectrum results shifted at lower BE and present a broadening increase of about 22%. This suggests an additional contribution of SiOx components to the defect peak. The SiO_2_ component at about 533.4 eV is strongly depressed.

O K edge XAS allows to figure out the structural variation in the O surrounding of the three samples. In the case of non- irradiated device, the spectrum agrees with literature results of SiO_2_ films [32].

After irradiation, the main edge feature is broadened, and an additional peak grows up. This feature at about 530 eV is assigned to the presence of defects, induced by neutron irradiation. The broadening of the main feature suggests a higher degree of disorder.

Annealing induces in the spectrum marked differences. In particular, the peak assigned to local defects in the films is strongly increased. An increased broadening takes place, confirming the presence of non-stoichiometric oxides in the coating film. We remind that in the FY the sampling depth exceeds the thickness of the coating layer.

In summary, the spectroscopic results imply that while irradiation induce local defects in the coating, the annealing induces non-stoichiometric oxides, in agreement with the results discussed in the case of Si2p spectra. The imaging by AFM supports these findings.

## 4. Discussion

In this work, Si L_2,3_ XAS and Si2p photoemission results provide new insights into the electrical behavior of a-Si:H devices after irradiation and subsequent annealing [11]. The strong increase of the Dis Si component, after irradiation, with the possible consequent increment of defects or voids, should cause the observed growth of leakage current and reduction of dosimetric sensitivity. Moreover, a reduction of the amount of Si-H bonds is found. Residual hydrogen trapped within the films cannot be revealed by the spectroscopies used in this work, though its presence is highly probable. The presence of residual interstitial hydrogen in a-Si:H films for example is found in Hydrogen effusion of a-Si:H films revealed as a function of annealing temperature [33]. After annealing, we observe an increase of the number of Si-H bonds, as can be seen looking at the core level measurements which together with the result obtained from electrical measurements [11] allows us to confirm the usefulness of the annealing.

Finally, we were able to characterize the effect of irradiation and subsequent annealing even in the coating layer. A higher degree of defects is detected in the irradiated sample, and an even higher degree of disorder of SiOx configurations is detected. These results could have implications for the device performance but electrical measurements were not taken during irradiation. Therefore, we can state that annealing process is able to partially compensate for the damage due to neutrons, as can be seen looking at the core level measurements which gain further support from AFM topography image Infrared and Raman Spectroscopy measurements are planned in the future to measure the density of Si-H bonds and the amount of interstitial Hydrogen to go further in the proposed model.

## 5. Conclusions

The mechanism of the radiation damage, first studied with a neutron source, and the effect of annealing on the devices after irradiation requires a multi-technique approach. In this work photoemission and XAS spectroscopies were applied to the characterization of devices formed by a-Si:H. It was possible to obtain information on the fingerprint of such material and to quantify the amount of Si-H bonds with respect to the amorphous Silicon network. The capability to observe on the device different types of aggregations, like c-Si-Si formations near the surface, and their evolution with depth, is a very important tool to understand the influence of production processes on the device properties.

The neutron irradiation induces an increase of the Dis Si component and a reduction of Si-H component by more than 50%. The subsequent annealing induces passivation of the created dandling bonds and a reduction of the dangling bonds density which is crucial for the device performance. This trend clearly explains the electrical behavior of the devices.

All observations made in this paper will be considered with a view to establish how they relate to the sensitivity of the device when exposed to the ionizing radiation.

The trend can be suggesting that before neutrons irradiation as well as after the annealing process we observe an higher concentration of Si-H bonds and better passivation of the dandling bonds, allowing for a higher carrier life time and carrier collection efficiency, in the same way a high concentration of a-Si dandling bonds, found in the irradiated sample, imply a quantity of deep defect with a high cross section of traps which leads to lower carrier life time and less efficient charge collection.

In this paper, as we have discussed neutron damage for the first time, we are unable to provide a direct comparison with other studies in the literature. However, as a future work we plan to study radiation damage with other sources: protons and photons. In this way we will be able to quantitatively compare the results obtained with the studies present in the literature.

Therefore, as the measurements are in agreement with what is obtained from the electrical measurements, this study provides an opportunity to understand more deeply how irradiation and annealing affect the performance of a-Si:H based devices.

## Figures and Tables

**Figure 1 nanomaterials-12-03466-f001:**
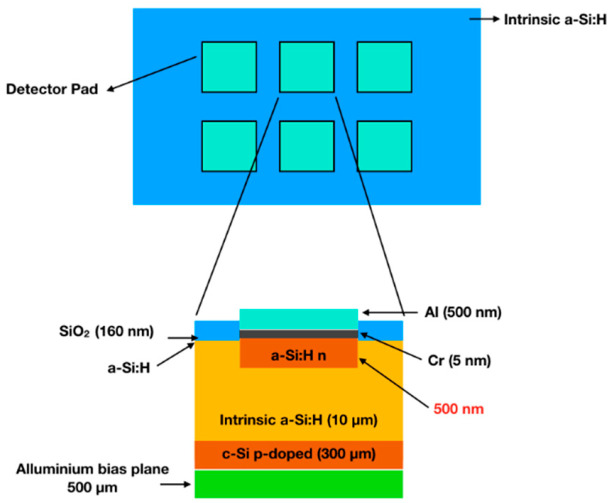
Top and side view of the p-i-n diode.

**Figure 2 nanomaterials-12-03466-f002:**
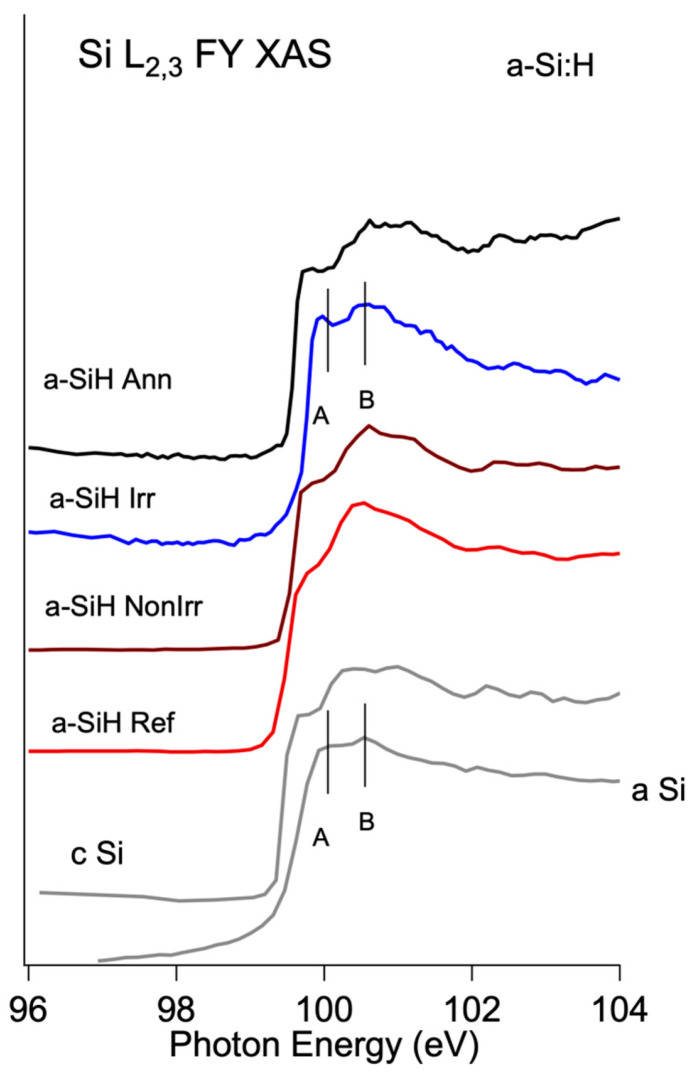
Si L_2,3_ XAS spectra for a-Si:H Ref and device samples measured in FY (black lines). Crystalline Si and amorphous Si spectra are shown for comparison (grey lines).

**Figure 3 nanomaterials-12-03466-f003:**
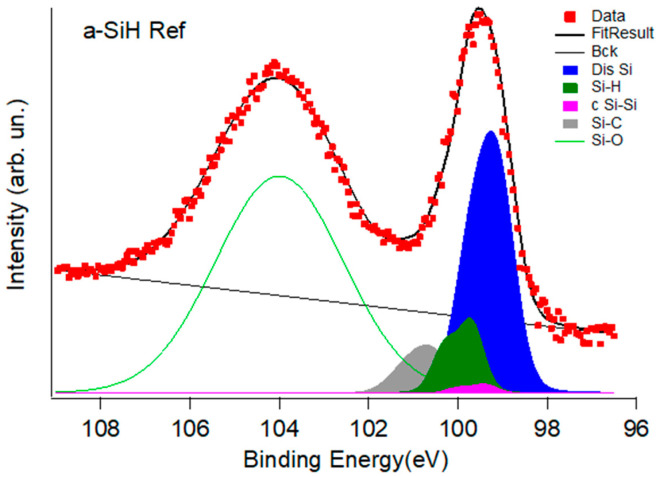
Si2p core level for the reference a-Si:H (a-Si:H Ref) sample measured at 1400 eV photon energy and the deconvolution into chemically shifted components (see also Table 1).

**Figure 4 nanomaterials-12-03466-f004:**
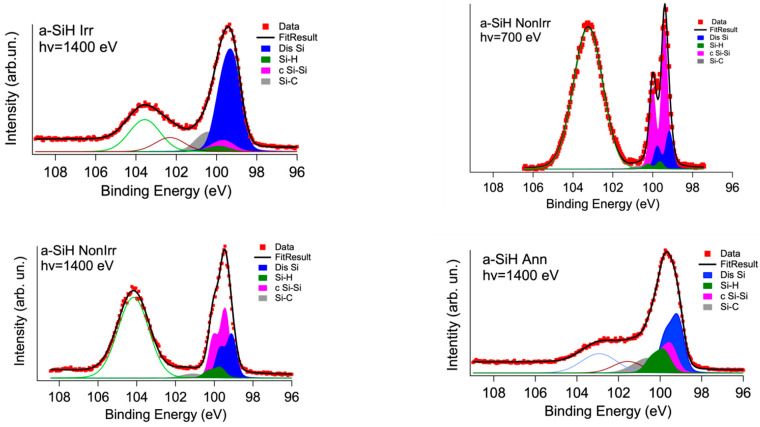
Si2p core level deconvolution for a-Si:H device samples measured in different samples: top left non-irradiated hv = 1400 eV, right non-irradiated hv = 700 eV; bottom left irradiated, bottom right: irradiated and annealed.

**Figure 5 nanomaterials-12-03466-f005:**
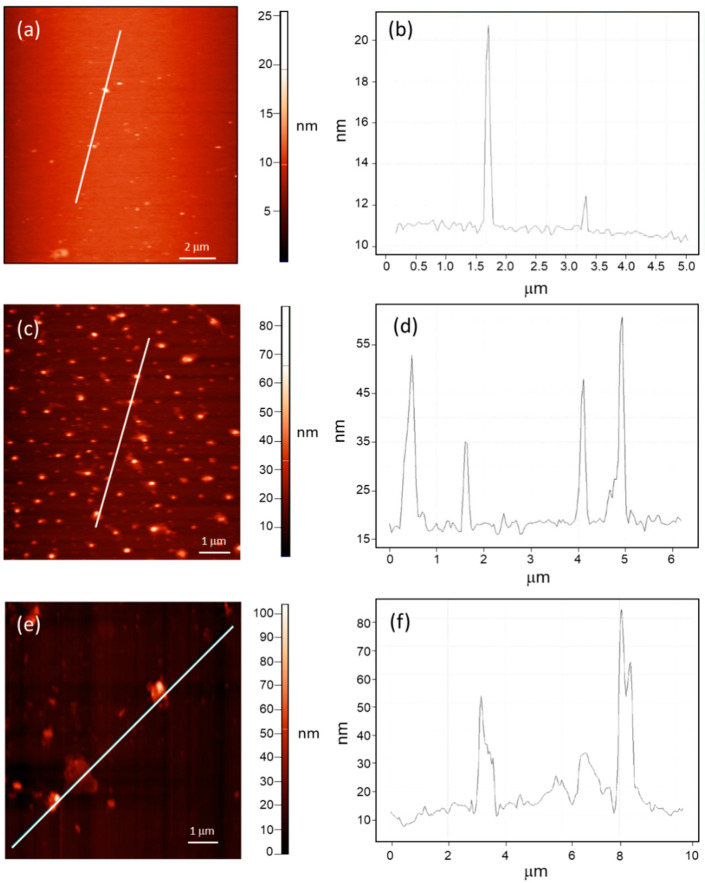
AFM topography image and height profile of the a-SiH NonIrr (**a**,**b**), a-SiH Irr (**c**,**d**), and the a-SiH Ann (**e**,**f**).

**Figure 6 nanomaterials-12-03466-f006:**
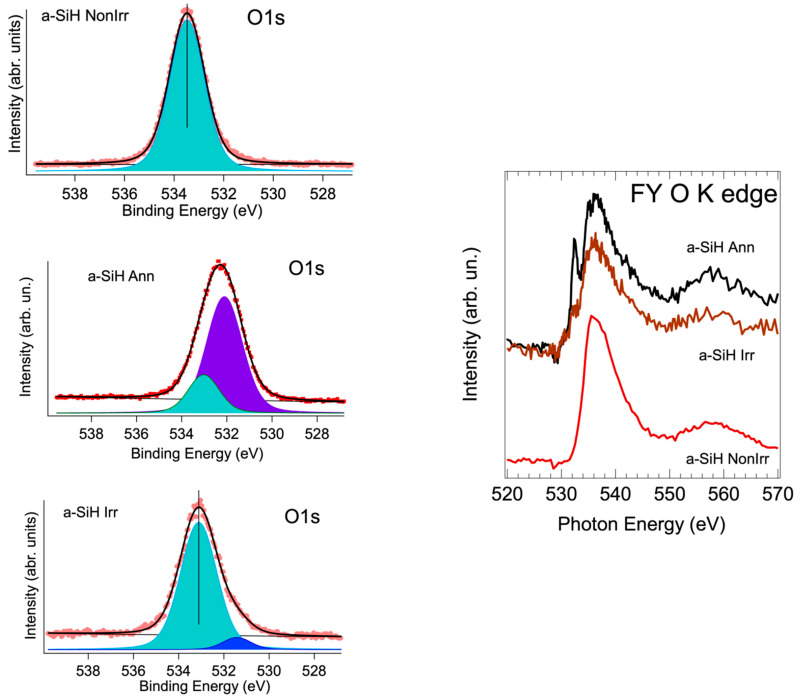
Left: O1s photoemission core level taken at 1400 eV for the three samples. The Non-irradiated sample is a single peak at 533.4 eV, as expected in SiO_2_ coating. Left: O K edge XAS spectra for the three samples.

**Table 1 nanomaterials-12-03466-t001:** Area of the Si2p components and Si2p_3/2_ BE, according to the deconvolutions shown in Figure 4. The concentrations are referred to as the Si2p components in the film (see text).

		a-Si:H Reference	No Irradiated	No Irradiated	Irradiated	Annealed
Component	Binding Energy (eV)	Concentration (1400 eV)	Concentration (1400 eV)	Concentration (700 eV)	Concentration (1400 eV)	Concentration (1400 eV)
c-Silicon (c Si-Si)	99.4 + 0.1	4%	38%	52%	17%	21%
a-Si & Dangling Bonds (Dis Si)	99.1 + 0.1	70%	46%	28%	67%	44%
Si-H	99.7 + 0.1	15%	11%	7%	3%	12%
Si-C	100.1 + 0.1	11%	5%	11%	13%	23%

## Data Availability

The data presented in this study are available in article.

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
