# Peer review of "High-Resolution Photoemission Study of Neutron-Induced Defects in Amorphous Hydrogenated Silicon Devices"

_nanomaterials, 2022, doi:10.3390/nano12193466_

Round 1

Reviewer 1 Report

This work investigates the radiation resistance of p-i-n devices based on hydrogenated amorphous silicon (a-Si: H) material by high-resolution photoemission, soft X-ray absorption and Atomic Force Microscopy. The amorphous silicon Si dangling bond passivation by hydrogen reduces their density by several orders of magnitude to reach a sufficiently low number of defects for usage in radiation detector devices. Furthermore, the damage induced by exposure to high energy irradiation of the material and its microscopic origin were investigated. A comparison of the spectroscopic results on bare and irradiated a-Si: H samples were also made well, showing that irradiation induces an increased degree of disorder and a strong reduction of the amount of the Si-H bonds. Further, the effects on the uppermost coating are also observed. This work tells a comprehensive story, but I still think the innovation is not strong, and the experiments are insufficient; thus, I don’t recommend publishing the work.

Following are some suggestions and basic questions that I found in the presented work:

  1. In Hydrogen passivation, where the dangling bond of Si reacts with hydrogen to produce a stable Si-H bond, it will certainly affect the radiation resistance of p-i-n devices based on hydrogenated amorphous silicon. Could the author enlighten this effect in their work?
  2. There are no references for reported numbers about concentration densities of the amorphous silicon in lines 77 and 79 of the main manuscript. Furthermore, the claim of bandgap incensement of a-Si with hydrogen reaction needs an authentic reference. Therefore, readers will be more informed if the authors provide solid references.
  3. The introduction part of this work needs proper attention to make it more meaningful, specifically from lines 87 to 99.
  4. If the authors redraw the schematic Figure.1 of the manuscript with consistent colours for each material, it will benefit the reader’s understanding of the work.
  5. Can the author elaborate on their statement about XPS spectroscopy measurement for the “real devices based on a-Si: H.” in line 113 ? or is it layers’ measurement separately?
  6.  In section 2, materials and methods, some abbreviation such as VHF, FBK etc., was presented without their complete names; therefore, it needs attention for clarity of the readers.
  7. The overall work lacks solid explanation and needs intense focus to modify.

Along with the above suggestions, the authors revised their work and presented some innovative results that can contribute effectively.

Reviewer 2 Report

Dear Editor,

In this manuscript, Peverini et al investigates the radiation resistance of p-i-n devices based on hydrogenated amorphous silicon by high-resolution photoemission, soft X-ray absorption and Atomic Force Microscopy. I think that the manuscript needs to be improved extensively before publication in Nanomaterials.

- Introduction does not reads nicely. I could not find enough background/context for the present study. I believe the a good introduction should briefly and firstly summarize the context, describe the issue that the authors are going to tackle and finally present how they actually tackled it. 

- Materials and methods: I think that I missed some important info here. The authors refer to [7] for the spectrum etc.. but I think they should put this info also here. What kind of source they used (spallation, reactor)? Is there any gamma-background that can contribute to the degradation of the device? I believe that these tests are important to test the device reliability in harsh radiation environments, such as outer space. Can the authors work our how many years of neutron irradiation in outer space (say on the ISS) correspond to their total dose?

Results and discussion: The annealing seems to worsen the radiation induced effects. Can the authors provide some explanation on this?

Reviewer 3 Report

Francesca Peverini et al. wrote this manuscript reports on the review paper about the radiation resistance of p-i-n devices based on hydrogenated amorphous silicon (a-Si: H) material by high-resolution photoemission. From my point of view, this manuscript contains information that can interest the scientific community, and I recommend its publication. However, amendments must be made before the final publication. Below are listed my observations.

1.  Typos and grammatical errors should be checked in the text.

2.  Please provide the TEM EELS elemental mapping to analyze the p-i-n device in the revised manuscript or supporting information.

3. How economical when it comes to production on an industrial level. Did the authors have any estimation?

4. The authors have constructed neutron-induced defects in amorphous hydrogenated silicon devices. Therefore, they need to compare the figures of merit with those obtained in the literature. Therefore, a comparison table in the revised manuscript is a necessity.

5.  Give and explain the function of Cr thin film in the p-i-n device.

Round 2

Reviewer 1 Report

Thanks authors for their efforts in manuscript revision. The biggest problem should be the novelty, it can't meet the requirement of this journal. English writing is still needed to be improved, and sentences should be wiritten in an academic way. I still can not convince myself to accept this version.

Reviewer 2 Report

Dear Editor, the authors have addressed most of my concerns.

Reviewer 3 Report

The authors have revised the manuscript according to the comments. Therefore, I recommend that the manuscript could be accepted for publication.